# A Comparative Analysis of Graphic Models for Enhancing Nutrition Education

**DOI:** 10.3390/nu17121947

**Published:** 2025-06-06

**Authors:** Magdalena Jodkiewicz, Justyna Malinowska, Karolina Marek-Woźny

**Affiliations:** Medical Center of Dietetics and Health Education, National Institute of Public Health NIH—National Research Institute, Chocimska 24, 00-791 Warsaw, Poland; jmalinowska@pzh.gov.pl (J.M.); kmarek-wozny@pzh.gov.pl (K.M.-W.)

**Keywords:** obesity, nutrition education, nutrition models, MyPlate, Healthy Eating Plate, prevention, eating habits

## Abstract

**Background**: Obesity is a global health problem associated with many chronic diseases. Nutrition education tools, such as graphic nutrition models, play a key role in the promotion of healthy eating habits. The purpose of this study was to compare selected nutrition models in plate form—MyPlate (USA), Harvard Healthy Eating Plate, the Eatwell Guide (UK), Malaysian Healthy Plate, and Polish Healthy Eating Plate—in terms of their structure, content, and effectiveness in improving diet quality and combating obesity. **Methods**: A comparative analysis of the selected models was carried out, focusing on their design, informational content, and compliance with dietary guidelines. A literature review was conducted in the PubMed and Embase databases. **Results**: The compared nutrition models are structurally similar but differ in the presence of additional information on the graphics, among other things. MyPlate and Malaysian Healthy Plate, compared to the other models, appear poor and lack recommendations on fats, beverages, and physical activity. The Eatwell Guide is distinguished by the proportion of each product group. Research indicates that plate models improve diet quality, but awareness of them in the public remains low. The Polish Healthy Eating Plate, while detailed, has not been empirically evaluated. **Conclusions**: Plate models are promising tools for nutrition education, but their effectiveness depends on adaptation to local conditions and ongoing educational efforts. It is necessary to conduct research evaluating the familiarity and effectiveness of the Polish Healthy Eating Plate in order to optimize the form of the message.

## 1. Introduction

Obesity is a chronic disease that does not resolve spontaneously and is prone to recurrence [1]. Its diagnosis is most commonly based on the Body Mass Index (BMI). Untreated obesity disease leads to the development of many other diseases and disorders, including cardiovascular disease, type II diabetes, metabolic disorders, and some malignancies [2]. Obesity also causes adverse economic effects and is the cause of 1.5 million hospitalizations in Poland, accounting for one-fifth of the health care budget [3]. Excessive body weight can also cause social stigma and consequently contribute to depression, anxiety disorders, sleep disorders, and even eating disorders [4].

Obesity represents a widespread public health issue in both Poland and globally. According to a 2022 report by the Supreme Audit Office, 65.6% of adult Poles were overweight or living with obesity [3]. In view of the above data, there is a need for intervention in the form of continuous nutrition education from an early age and the creation of appropriate tools in the form of dietary models.

Modern dietary recommendations in European countries are presented in a variety of graphic forms that reflect both current scientific knowledge and the cultural conditions of individual societies. Regardless of the form of graphic representation of healthy eating recommendations, all countries regularly update their dietary recommendations to improve population health and reduce the incidence of obesity and chronic diseases [5]. In recent years, there has been a global trend away from traditional pyramids to more intuitive plate models, which was initiated by the United States Department of Agriculture (USDA) with the publication of the MyPlate model in 2011 (Figure 1) [6]. It is noteworthy that different countries are adapting it to local eating habits and available foods [5]. This change is particularly well illustrated by the example of Poland, which, like other European countries, for many years used the traditional food pyramid. In 2020, the National Institute of Public Health—National Institute of Hygiene introduced a new model, the Healthy Eating Plate, which represented a major change in nutrition education [7].

European dietary guidelines, despite the common goal of promoting health, vary from country to country, taking into account local culinary traditions, product availability, and the specific health needs of the population. In Europe, countries such as Romania, Bulgaria, Cyprus, Slovenia, Estonia, Ireland, Lithuania, Luxembourg, Austria, Finland, and Switzerland continue to promote the pyramid model. Perhaps this is due to the strong entrenchment of the Mediterranean diet concept in the public mind. In contrast, European countries such as the United Kingdom, Germany, Portugal, Spain, Croatia, Latvia, Hungary, Malta, the Netherlands, Slovakia, and Iceland have adopted the plate form, which more directly depicts the ideal proportions of meal ingredients. Some countries, such as Sweden, France, the Czech Republic, Greece, and Denmark, have their own proprietary graphic models and written recommendations. Among other things, they include information on what to eat less and more of, general nutrition advice, or graphics promoting a plant-based diet [5].

The transformation of the Polish dietary model from the pyramid [8] to the plate [9] reflects a broader European trend toward the simplification and practical adaptation of nutritional guidelines [5]. This shift not only enhances the communication of dietary knowledge to the general public but also more effectively responds to contemporary challenges in nutritional prevention, particularly amid the escalating epidemic of obesity and lifestyle-related diseases [10]. Future research should prioritize evaluating the effectiveness of the plate model in shaping healthy eating habits. It should also consider the variations among existing visual representations in order to develop a format that is both accessible and practical for diverse target groups, thereby improving the overall impact of nutritional education. Therefore, the purpose of this study was to compare scientifically evaluated models of dietary recommendations from different countries with the Polish nutrition model in order to identify similarities and differences and potential areas for modification, and to propose directions for improvement based on research results.

## 2. Materials and Methods

### 2.1. Literature Review for Selection of Graphic Models for Analysis

We conducted a comparative analysis of selected nutrition models: MyPlate, Harvard Healthy Eating Plate, the Eatwell Guide, Malaysian Healthy Plate, and Polish Healthy Eating Plate. Additionally, we searched the PubMed and Embase databases and selected articles on dietary recommendations presented in the plate form using the following terms: “nutrition plate models”, “MyPlate”, “Healthy Eating Plate”, “nutrition education”, “the Eatwell Guide”, and “obesity prevention”.

Peer-reviewed scientific articles, government publications, and dietary guidelines from 2011–2024 were included. The selection of the dietary models analyzed was based on criteria including: (1) the availability of a sufficient number of scientific publications evaluating their effectiveness, (2) the reference model’s status in the literature-in the case of the US model as the first globally implemented plate system, and (3) official implementation in national nutrition education systems.

Justification:MyPlate (USA, 2011)—the first nationwide plate model, serving as a reference for later studies.Healthy Eating Plate (Harvard, 2012)—a model with proven effectiveness, widely cited in the literature.The UK’s the Eatwell Guide (UK, 2016)—well described in the literature both in terms of structure and has scientific research.Malaysian Healthy Plate (Malaysia, 2016)—well described in the literature both in terms of structure and has scientific research.Healthy Nutrition Plate (Poland, 2020)—representative of regional guidelines, included due to the timeliness of implementation and lack of comprehensive assessments in the literature.

Inclusion criteria:Articles and models published in English or Polish.Studies evaluating the effectiveness, awareness, or implementation of dietary recommendations in the plate format.Data on cultural, economic, or educational barriers to the use of the models.

The literature review was conducted by two experts. Initially, 29 literature items found by keywords that matched the topic of the paper and complied with the inclusion criteria were included in the review. After careful analysis, 2 publications were rejected due to inconsistency with the others.

### 2.2. Qualitative Analysis of Graphic Models

Descriptive analysis was used to compare the models, with a focus on structural differences and the amount of descriptive data included. In light of the absence of prior studies comparing the models, a comprehensive evaluation of their graphical and descriptive components was conducted, with the aim of identifying similarities and differences and contextualizing these findings within the literature on recommendation effectiveness.

Limitations:-A lack of studies on the Polish Healthy Eating Plate.-A variety of methodologies and measured outcomes in the analyzed literature.

## 3. Results

Dietary recommendations presented in the plate format are generally similar across countries; however, they differ in aspects such as the inclusion of supplementary information within the graphic, particularly regarding the quality of the recommended food products. According to the Nazmi et al. study, nutrition education that includes additional information on the degree of food processing, in this case NOVA, is more effective than using only the MyPlate model [11]. Some plate models include such information, such as Harvard’s Healthy Eating Plate, the Eatwell Guide, or the Polish Healthy Eating Plate, but they differ in their amount and form of presentation [7,12,13]. Researchers in the United Kingdom have observed that while the Eatwell Guide offers recommendations for maintaining a healthy and balanced diet, it does not address the role of ultra-processed foods. As a result, it remains uncertain whether a diet largely composed of such foods can still be considered healthy, thereby underscoring the need for further empirical investigation [14].

The MyPlate (Figure 1), Harvard Healthy Eating Plate (Figure 2), Malaysian Healthy Plate (Figure 3), and polish Healthy Eating Plate (Figure 4a,b) models are similar in terms of the recommended high intake of fruits and vegetables—this group in all these models occupies half of the plate volume. Additionally, Harvard Healthy Eating Plate, Malaysian Healthy Plate, and Polish Healthy Eating Plate carbohydrate products take up around 25% of the meal, the same as protein products. MyPlate’s similarity to Harvard’s Healthy Eating Plate ends with the graphical presentation. MyPlate is a simpler model, with no in-depth guidance on the quality of the cereal or protein products chosen, and the element of fat addition to meals is absent. Instead, the model is distinguished by the recommendation to consume milk and dairy products—implicitly liquid as a beverage—with meals. Malaysian Healthy Plate is a simple model too, in which, however, attention is paid to the diversity and quality of sources of complex carbohydrates and recommended sources of protein. On the other hand, the issue of beverages and the addition of fat to meals was omitted.

The Eatwell Guide (Figure 5) is distinguished by the recommended proportions in meals (over 1/3 of the volume occupied by vegetables and fruits, the same amount by carbohydrate products, 25% by protein products, and 5% by fats). Just like the Harvard Healthy Eating Plate and the Polish Heathy Eating Plate, it includes the recommendation to consume water, teas, and coffee, and the addition of oil to meals. However, it lacks the recommendation of being physically active, which the two other have. Harvard and Polish Healthy Eating Plate differ in their message regarding dairy, as the Harvard model recommends limiting the intake of milk and dairy products to 1–2 servings per day, while in the Polish one dairy is included as one of the recommended sources of protein, and even a recommendation is made to increase the intake of low-fat dairy products, especially fermented ones. The Polish Heathy Eating Plate is distinguished by the volume of text and the addition of graphics with the “3 Steps to Health!” healthy eating recommendations (Figure 4b). The models are compared in Table 1.

**Figure 1 nutrients-17-01947-f001:**
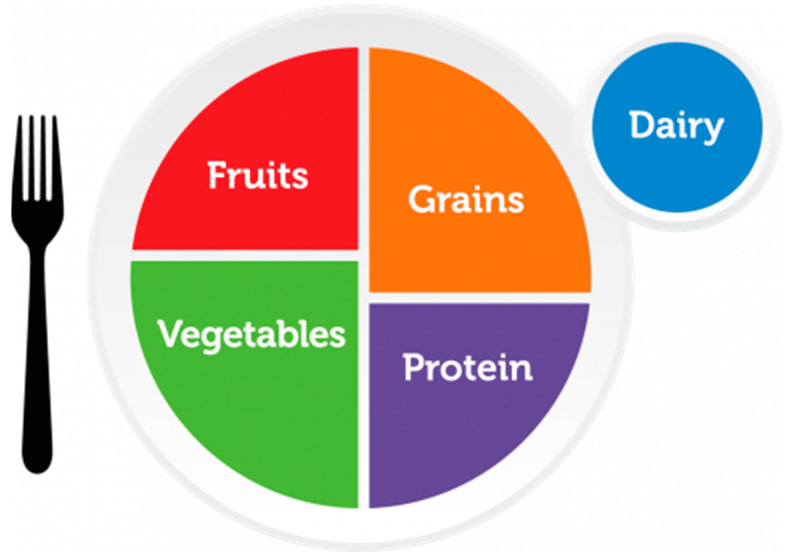
MyPlate [15].

**Figure 2 nutrients-17-01947-f002:**
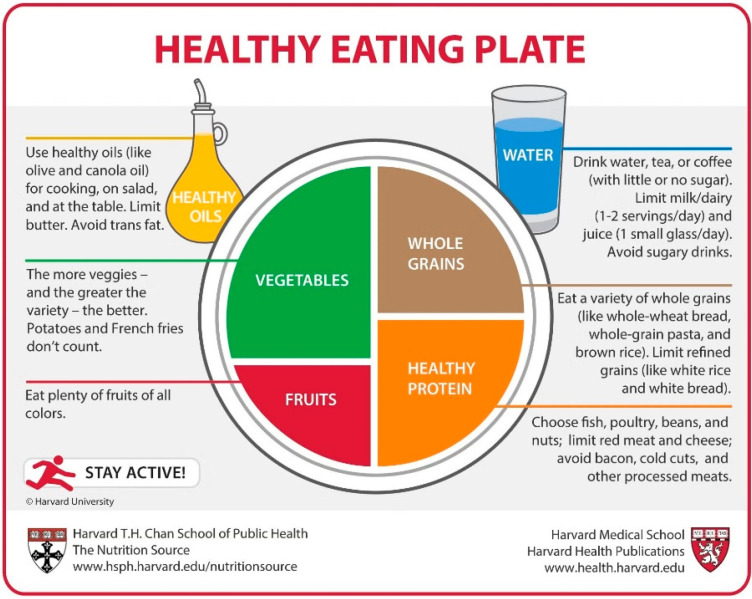
Harvard Healthy Eating Plate. Copyright© 2011, Harvard University. For more information about The Healthy Eating Plate, please see The Nutrition Source, Department of Nutrition, Harvard T.H. Chan School of Public Health, *www.thenutritionsource.org*; and Harvard Health Publications, *www.health.harvard.edu* (accessed on 27 March 2025) [16].

**Figure 3 nutrients-17-01947-f003:**
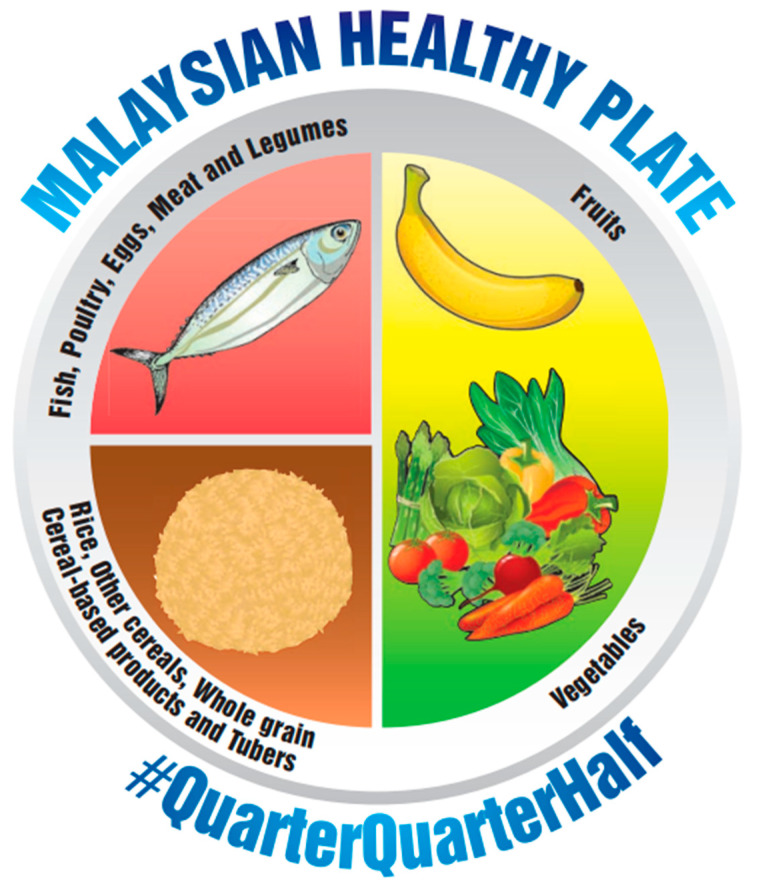
Malaysian Healthy Plate [17].

**Figure 4 nutrients-17-01947-f004:**
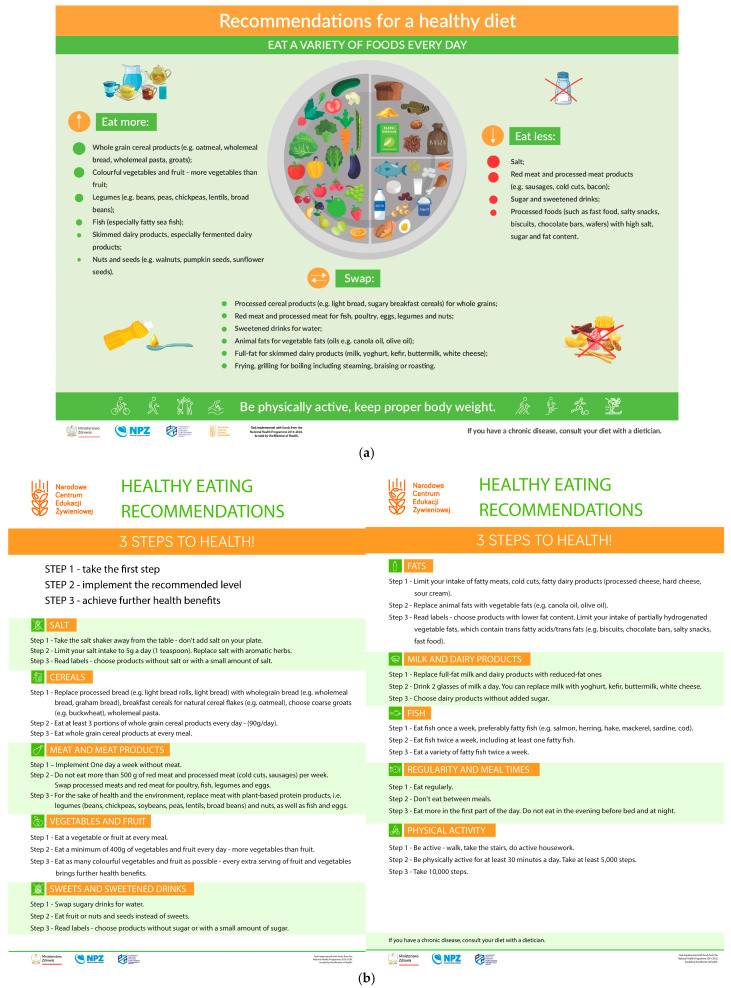
(**a**) Polish Healthy Eating Plate. (**b**) Healthy Eating Recommendations—3 Steps to Health! [9].

**Figure 5 nutrients-17-01947-f005:**
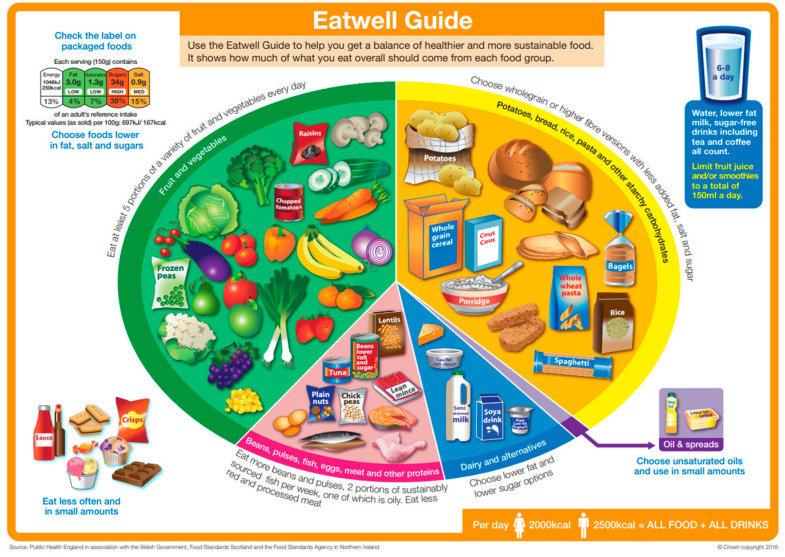
The Eatwell Guide [13].

To summarize the above comparison, the first aspect to be taken into account in a possible update of the model is the legibility of the depicted division of the plate into parts. Overly small differences between the different parts make it difficult to apply the presented proportions in practice. This is particularly evident when comparing the MyPlate and Harvard Healthy Eating Plate models. In addition, the comparison notes a significant disparity in the amount of information in MyPlate and Malaysian Healthy Plate compared to the Harvard Healthy Eating Plate, the Eatwell Guide, and the Polish one. In this comparison, MyPlate and Malaysian Healthy Plate appear poor and clearly lack information on fat, beverage, and physical activity recommendations. On the other hand, the Polish Healthy Eating Plate may appear overly detailed, yet it offers comprehensive information on healthy eating principles and does not overlook controversial dietary elements such as sweets, salty snacks, and salt.

Most nutrition models, such as the Harvard Healthy Eating Plate, MyPlate, or Polish Healthy Eating Plate models, do not include potatoes in their structure in any food group. The exception is the Eatwell Guide, which specifically lists potatoes in the starchy carbohydrate category. The Malaysian Healthy Plate does not indicate potatoes but includes tubers in general in the carbohydrate food group. The lack of consistent coverage of potatoes in international dietary recommendations can lead to ambiguity in their classification and consumption recommendations. A comparative analysis of the dietary models also noted that nuts are included as a protein food in the Eatwell Guide, Harvard Healthy Eating Plate, and Polish Healthy Eating Plate, while they are not listed in the Malaysian Healthy Plate and MyPlate models. Given their high fat content, it is worth considering whether they should be classified primarily as a source of vegetable fats, which could contribute to more consistent dietary recommendations.

## 4. Discussion

Nutrition models such as MyPlate in the US, Harvard Healthy Eating Plate, the Eatwell Guide in the United Kingdom, the Malaysian Health Plate, and the Polish Healthy Eating Plate are important educational tools to improve eating habits. Although the number of studies evaluating their effectiveness is very limited, the results are promising. In the National Health and Nutrition Examination Survey (NHANES) conducted on many thousands of Americans, it is shown that the use of MyPlate is associated with healthier eating habits and better diet quality in American adults [10,18]. Moreover, among patients with type 2 diabetes, MyPlate-based education significantly improved glycemic control and lipid profile. The results showed statistically significant reductions in serum fasting glucose (*p* = 0.014) and glycated hemoglobin (*p* < 0.001), as well as a reduction in low-density lipoprotein (*p* = 0.043) [19]. MyPlate may also be a practical alternative to traditional calorie counting to promote a feeling of satiety and facilitate the reduction of visceral fatty tissue, as confirmed in a randomized trial involving 261 Americans [20]. A study conducted by Gregory and colleagues involving 517 participants demonstrated that greater adherence to the recommendations of the Eatwell Guide was significantly associated with lower blood pressure and a reduced body mass index. These findings suggest that the Eatwell Guide may indirectly contribute to a reduced risk of dementia by improving cardiometabolic parameters [21].

The broader significance of the Eatwell Guide for public health and sustainable development was discussed in a report by Shannon et al., which summarized the outcomes of an expert forum held in 2023. The report emphasizes that although the evidence supporting the benefits of the Eatwell Guide is limited, there are clear indications of its positive impact on both population health (e.g., reduced all-cause mortality, decreased abdominal obesity in postmenopausal women) and the environment (e.g., reduced greenhouse gas emissions) [22]. Similar conclusions were drawn by Scheelbeek et al., who analyzed the impact of the Eatwell Guide on mortality. Their findings suggest significant health benefits and a reduced risk of mortality among individuals adhering to the guide’s recommendations [23]. Comparable results were reported by Fadnes et al., who highlighted that following healthy dietary patterns may prevent the development of non-communicable diseases and positively influence life expectancy. Moreover, a sustained dietary shift from unhealthy patterns to those aligned with the Eatwell Guide was associated with an increase in life expectancy of 8.9 years for 40-year-old men and 8.6 years for women of the same age [24].

Despite these positive effects, awareness of such models in the general public remains low. In one NHANES study involving 1693 American adults with young children, only 29% had heard of MyPlate, and among 3521 eighth-grade students, only 11% correctly answered questions about the tool [25,26]. The situation is even worse for the Malaysian Health Plate, where researchers at the National Institute of Health estimate that nearly 80% (16.9 million) of adults have never heard of the model [27]. Similar problems are observed in Saudi Arabia, where very low adherence to the Saudi Healthy Plate recommendations was reported among 1092 female participants. Only 0.7% of the participants in the study adhered to the recommendations and had no symptoms of eating disorders, and those who had eating disorders had more than 50% poor adherence to the Saudi Healthy Plate recommendations [28].

Recent analyses of the 2013–2018 NHANES survey of more than 17,000 participants also found that public awareness of MyPlate among American adults is low but growing. The study noted significant differences in awareness of dietary recommendations by socioeconomic status. Those receiving assistance programs showed a more rapid increase in awareness, which may indicate the effectiveness of educational efforts in these groups [29]. These assumptions also support previous observations by Wambogo et al., who noted that familiarity with MyPlate correlates with better self-assessment of diet quality [30].

In terms of the practical application of the plate model, Wansink and Kranz’s study based on a national online survey, to which 497 mothers responded, revealed that early users of MyPlate were mainly mothers with nutritional knowledge and cooking experience who appreciated the transparency of the tool. This underscores the importance of basic nutrition education for the successful implementation of such models [31]. Another randomized trial involving 160 pairs (parent–child) showed that the half-plate rule filled with fruits and vegetables with MyPlate can be an effective tool in forming healthy habits. Parents using this recommendation were more likely to report positive eating practices at home, suggesting that simple, visual messages can be particularly effective in family nutrition education [32].

In the context of the diversity of dietary patterns, it is worth noting that studies comparing different dietary patterns have shown interesting relationships. Turner-McGrievy et al., in a randomized trial involving 227 African-Americans, showed that all three dietary patterns recommended in the US Dietary Guidelines, i.e., Healthy US-Style, the Mediterranean diet, and the vegetarian diet, when followed using MyPlate principles, had similar weight reduction benefits. This suggests that the MyPlate structure itself may be a universal basis for different dietary approaches [33]. Similar conclusions were reached by Best and Flannery in a study involving 4162 postmenopausal women, the results of which suggest that adherence to either the Mediterranean diet or the Eatwell Guide may help prevent abdominal obesity in postmenopausal women [34].

Barriers to the use of these models are complex and include cultural, economic, and educational factors. For example, in a study involving more than 300 Asians in the U.S., dietary preferences such as eating only refined grains (mainly white rice) affected 30.7% of respondents, making it difficult to use MyPlate. In the same study, 22.4% of Asians did not eat half their plate of fruits and vegetables [35]. In Tanzania, restrictions were mainly related to the availability of healthy foods from street vendors [36]. In U.S. schools, teachers cited lack of time and resources as factors preventing effective integration of MyPlate into curricula [37].

Findings from this research point to the need to adapt nutrition tools to local cultural conditions and to intensify educational efforts. Simple nutrition graphics, such as plate models, can increase our understanding of proper nutrition [38]. Nutrition models, as a tool for nutrition education, have the potential to improve people’s diets and health, but their effectiveness depends on a comprehensive approach that includes public awareness, education, and system support.

## 5. Conclusions

Obesity is a major health, social, and economic challenge both in Poland and around the world. In response to this growing problem, countries are introducing a variety of dietary models to educate the public and promote healthy eating habits. In recent years, there has been a global trend toward replacing food pyramids with more intuitive plate models, such as MyPlate, Harvard Healthy Eating Plate, the Eatwell Guide, Malaysian Healthy Plate, and Polish Healthy Eating Plate. The available literature data show that plate models can effectively improve diet quality and support the fight against obesity, but their awareness among the public remains low. It is important to adapt these tools to local cultural conditions and to intensify educational efforts, especially in groups of lower socioeconomic status. The literature data show that there are still few works assessing the familiarity, degree of use, and effectiveness of existing plate models. In the case of the Polish Healthy Eating Plate, there are no such studies at all. There is a need for such analyses, and for this reason studies are planned that will focus on evaluating the effectiveness of the models in forming eating habits, which will allow for the optimization of the form of the message and the updating of the nutrition model.

## Figures and Tables

**Table 1 nutrients-17-01947-t001:** Comparison of the MyPlate, Harvard Healthy Eating Plate, Eatwell Guide, Malaysian Healthy Plate, and Polish Healthy Eating Plate models.

Parts of Model	MyPlate	Harvard Healthy Eating Plate	The Eatwell Guide	Malaysian Healthy Plate	Polish Healthy Eating Plate
Vegetables and fruits	Vegetables and fruits take up a total of 1/2 the volume of the plate.	Vegetables and fruits take up a total of 1/2 the volume of the plate.	Vegetables and fruits take up over 1/3 the volume of the plate.	Vegetables and fruits take up a total of 1/2 the volume of the plate.	Vegetables and fruits take up a total of 1/2 the volume of the plate.
	There is a clear division within this group, i.e., about 60% vegetables and 40% fruits.	There is a clear division within this group, i.e., about 70% vegetables and 30% fruits.	There is no clear division within this group; however, there are more vegetables than fruits.	There is no clear division within this group, but about 60% of volume is taken up by vegetables and 40% by fruits.	There is no clear division within this group.
		Vegetables should be varied and fruits should be colorful; potatoes and French fries are not included in this group.	It is recommended to eat at least five portions of a variety of fruit and vegetables every day.		There should be more vegetables than fruits, and they should be of different colors.
Starchy carbohydrates	Cereals occupy 30% of the plate volume.	Cereals occupy 25% of the volume of the plate.	Potatoes, bread, pasta, or other starchy carbohydrates occupy around 35% of the volume of the plate.	Rice, other cereals, whole grain cereal-based products and Tubers occupy 25% of the volume of the plate	Cereals occupy 25% of the volume of the plate.
	Half of the cereal products consumed should be whole grains.	All grain products should be whole grains, preferably a variety.	Grain products should be higher-fiber versions or wholegrain, and potatoes should be eaten with skin.		All grain products should be whole grains.
Protein products	Protein products take up 20% of the volume of the plate.	Protein products take up 25% of the volume of the plate.	Protein products take up approximately 25% of the volume of the plate (15% beans, pulses, fish, eggs, meat, and other protein foods, and 10% dairy or dairy alternatives).	Fish, poultry, eggs, meat, and legume take up 25% of the volume of the plate.	Protein products take up 25% of the volume of the plate.
	They should be diverse.	Fish, poultry, beans, and nuts are most recommended; red meat and cheese should be limited, and bacon and cold cuts avoided.	Choosing lean cuts of meat and mince is recommended; red and processed meat should be limited; and lower-fat and lower-sugar dairy products are recommended. Fish should be eaten twice per week, of which at least one portion should be oily fish.		Poultry, fish, eggs, milk and dairy products, pulses, and nuts are the most recommended; it is recommended to limit red meat and processed meat products.
Fats	They have not been taken into account.	A bottle of oil is next to the plate.	Fats take up approximately 5% of the volume of the plate. Unsaturated fats like vegetable, rapeseed, olive, and sunflower oils are recommended.	They have not been taken into account.	A bottle of oil is next to the plate, along with a spoon used to measure portions of fat.
		Moderate fat intake is recommended; the recommended fats are olive oil and canola oil, while butter should be limited and trans fats avoided.			A small addition of fat to the meal is recommended; it is advisable to replace animal fats with vegetable fats (oils, e.g., canola, olive oil).
Beverages	Drinking milk or dairy products with each meal is recommended.	Next to the plate is a glass of water. It is recommended to drink water, tea, or coffee (with little or no sugar); to limit the consumption of milk and dairy products to 1–2 servings per day and juices to one small glass per day; and to avoid sweetened beverages.	Next to the plate is a glass of water. It is recommended to drink water; lower-fat milk; and sugar-free drinks, including tea and coffee, and to limit the consumption of juices and/or smoothies to a total of 150 mL a day.	They have not been taken into account.	Some examples of recommended beverages (water, tea, and coffee) are next to the plate.
Other	No	No	A group of sweets and salty snacks is next to the plate, with a recommendation to eat them less often and in small amounts. An example of a label is next to the plate, with a recommendation to check labels on packaged foods and to choose foods lower in fat, salt, and sugars. Information on recommended daily energy intake for females (2000 kcal) and males (2500 kcal) from all foods and all drinks is at the bottom.	No	A crossed-out salt shaker is next to the plate. A crossed-out group of sweets and salty snacks is next to the plate.
Physical activity recommendation	No	The graphic is complemented by the recommendation “Stay active!’’.	No	No	The graphic is complemented by the recommendation “Be physically active, keep body weight normal”.

## Data Availability

The original contributions presented in this study are included in the article. Further inquiries can be directed to the corresponding author.

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
