# Peer review of "A Comparative Analysis of Graphic Models for Enhancing Nutrition Education"

_nutrients, 2025, doi:10.3390/nu17121947_

Round 1

Reviewer 1 Report

Comments and Suggestions for Authors

Thank you very much for allowing me to review the article entitled “Comparative analysis of selected graphic models as a tool for nutrition education in the world with an assessment of their effectiveness in the context of the global obesity epidemic.” While the topic is of interest to professionals in the field of public nutrition, there are several aspects that do not allow me to approve this version of the manuscript.

  1. I don't understand why the manuscript is classified as a review. It has more characteristics of an original article.
  2. The introduction does not state the study's objective or why it is necessary.
  3. The authors should be careful with their language. Replace 'obese' with 'people with obesity' or 'people living with obesity.'
  4. There is no clear explanation for why only images with a plate format were selected.
  5. There are some paragraphs that appear to be unfinished. For example, on line 61, '(source)' appears in parentheses. It seems there is a missing reference there. See Table 1; there is a missing column title there.
  6. How was the comparison performed? How were the comparison criteria selected? Who did the comparison?
  7. It should be considered that the images complement the messages of the dietary guidelines; therefore, it is difficult to separate the effect of one from the other.
Comments on the Quality of English Language

I am not able to make comments about the English

Author Response

Dear Reviewer,

Thank you very much for taking the time to review this manuscript. Please find the detailed responses below and the corresponding revisions highlighted in the re-submitted files.

  1. I don't understand why the manuscript is classified as a review. It has more characteristics of an original article.

Thank you for your suggestion. The manuscript has been changed from a review to an original article.

  1. The introduction does not state the study's objective or why it is necessary.

Thank you for your remark. The purpose of the study was added to the introduction: “Therefore, the purpose of this study was to compare scientifically evaluated models of dietary recommendations from different countries with the Polish nutrition model in order to identify similarities and differences and potential areas for modification, and to propose directions for improvement based on research results.”

  1. The authors should be careful with their language. Replace 'obese' with 'people with obesity' or 'people living with obesity.'

Thank you very much for pointing this out. The text no longer contains phrases such as “obese people”.

  1. There is no clear explanation for why only images with a plate format were selected.

Thank you for bringing this issue to our attention. The answer to this comment can be found in the introduction of the article, where it is explained that the choice of plate format is due to the global trend away from traditional food pyramids to more intuitive plate models. This trend was started by the United States Department of Agriculture (USDA) with the publication of the MyPlate model in 2011. Additionally, the purpose of this study was to compare scientifically evaluated models of dietary recommendations from different countries with the Polish nutrition model in order to identify similarities and differences and potential areas for modification, and to propose directions for improvement based on research results.

  1. There are some paragraphs that appear to be unfinished. For example, on line 61, '(source)' appears in parentheses. It seems there is a missing reference there. See Table 1; there is a missing column title there.

Thanks for pointing out the errors. They have been corrected.

  1. How was the comparison performed? How were the comparison criteria selected? Who did the comparison?

We have reclassified the manuscript as an original article and added a Materials and Methods section where this is described.

  1. It should be considered that the images complement the messages of the dietary guidelines; therefore, it is difficult to separate the effect of one from the other.

Thank you for your feedback. In our manuscript, we focused on evaluating the tools that are used during nutrition education, rather than the dietary guidelines that often complement the nutrition model's message. Available scientific publications have evaluated the effectiveness of nutrition models only, which gave direction to our study.

Best regards

Authors

Reviewer 2 Report

Comments and Suggestions for Authors

The manuscript proposes a review on the potential effectiveness of qualitative cues representing in the form of graphic models for guidance in nutrition education.

However, the paper requires extensive revision to improve its contents.

First, the title is too long and includes assertions that may be either very controversial or too ambitious in the context of a very short review article (i.e., how are authors supposed to perform a brief review on the graphic models "with an assessment of their effectiveness in the context of the global obesity epidemic"?).

Second, the Introduction of the study is quite superficial, including sentences using informal language (e.g., "Obesity is a chronic disease , which does not go away on its own and tends to recur"; "The problem of obesity is prevalent both in Poland and around the world", among others). A proper review on the use of graphical models and/or qualitative approaches of dietary guidelines for nutritional education is required to improve the Introduction.

Third, the study mentions several controversial approaches (e.g., NOVA) and evaluations of graphical models and/or qualitative approaches of dietary guidelines (e.g., studies published by Brian Wansink, a champion of retracted studies in the field of marketing applied to food and nutrition), lacking critical assessment on the literature cited, including recent publications with criticisms on the variability of classifications used to identify "ultra-processed" foods. Furthermore, these elements do not contribute to the context or the analyses conducted in the study, therefore, could be eliminated without losses to the investigation presented by the authors.

Fourth, the analysis provided in the context of graphic models is quite interesting (Figures 1-3 and Table 1); however, the methods for selection of models, the analyses performed, and the results presented are only briefly described in the manuscript. Authors should convert the manuscript into an original investigation, building the study around the analyses proposed of the graphic models. In addition, including other graphic models could enrich the manuscript, creating interest for readers in the field of knowledge through the expansion of the implications of the analyses conducted into public policies of nutrition and health.

Finally, the Discussion and the Conclusions should be improved to explore the implications previously mentioned, allowing to explore additional uses of graphic models beyond the superficial description presented in the text.

Comments on the Quality of English Language

The quality of English language requires major improvements, particularly in the use of informal language.

Author Response

Dear Reviewer,

Thank you very much for taking the time to review this manuscript. Please find the detailed responses below and the corresponding revisions highlighted in the re-submitted files.

1. First, the title is too long and includes assertions that may be either very controversial or too ambitious in the context of a very short review article (i.e., how are authors supposed to perform a brief review on the graphic models "with an assessment of their effectiveness in the context of the global obesity epidemic"?).

Thank you for your remark. The title has been simplified and is consistent with the content of the article.

2. Second, the Introduction of the study is quite superficial, including sentences using informal language (e.g., "Obesity is a chronic disease , which does not go away on its own and tends to recur"; "The problem of obesity is prevalent both in Poland and around the world", among others). A proper review on the use of graphical models and/or qualitative approaches of dietary guidelines for nutritional education is required to improve the Introduction.

Thank you for your comment. The introduction in our article was to explain why nutrition education is essential and to outline changes and global trends in the design of nutrition education tools. The introduction of the article explaines that the choice of plate format is due to the global trend away from traditional food pyramids to more intuitive plate models. This trend was started by the United States Department of Agriculture (USDA) with the publication of the MyPlate model in 2011. Additionally, the purpose of this study was to compare scientifically evaluated models of dietary recommendations from different countries with the Polish nutrition model in order to identify similarities and differences and potential areas for modification, and to propose directions for improvement based on research results.

In accordance with the reviewers' suggestions, the character of the article was changed to an original article. Accordingly, the suggested review on the use of graphical models and/or qualitative approaches of dietary guidelines for nutritional education is in discussion section.

3. Third, the study mentions several controversial approaches (e.g., NOVA) and evaluations of graphical models and/or qualitative approaches of dietary guidelines (e.g., studies published by Brian Wansink, a champion of retracted studies in the field of marketing applied to food and nutrition), lacking critical assessment on the literature cited, including recent publications with criticisms on the variability of classifications used to identify "ultra-processed" foods. Furthermore, these elements do not contribute to the context or the analyses conducted in the study, therefore, could be eliminated without losses to the investigation presented by the authors.

Thank you for your feedback. In our article, we used studies evaluating the effectiveness of tools used for nutrition education that take into account the quality and degree of food processing to point out that supplementing education with this aspect can contribute to better dietary choices for people. The issue of the degree of food processing is important to address because of the increasing prevalence of ultra-processed products on the market, not because of the results of this particular study.

4.Fourth, the analysis provided in the context of graphic models is quite interesting (Figures 1-3 and Table 1); however, the methods for selection of models, the analyses performed, and the results presented are only briefly described in the manuscript. Authors should convert the manuscript into an original investigation, building the study around the analyses proposed of the graphic models. In addition, including other graphic models could enrich the manuscript, creating interest for readers in the field of knowledge through the expansion of the implications of the analyses conducted into public policies of nutrition and health.

Thank you for your suggestion. The manuscript has been changed from a review to an original article. In addition, the article includes other graphical models for comparative analysis.

5. Finally, the Discussion and the Conclusions should be improved to explore the implications previously mentioned, allowing to explore additional uses of graphic models beyond the superficial description presented in the text.

The discussion has been refined and expanded to take into account the nutritional models being compared. The database of articles on the use of the models is limited.

Best regards

Authors

Reviewer 3 Report

Comments and Suggestions for Authors

This review manuscript focuses on obesity and points out that in response to the growing problem, countries are introducing various dietary patterns to educate the public and promote healthy habits. Although in recent years, intuitive plate models have been used to replace the food pyramid. However, this manuscript only cites the United States' "My Plate", Harvard University's "Healthy Eating Plate" and Poland's "Healthy Eating Plate". Is there no relevant information for reference or comparison in other countries?

The United States launched "My Plate" and Harvard University's "Healthy Eating Plate" not only to improve the public's obesity problem, but also to consider other aspects of healthy eating, nutritional balance and disease prevention. It is recommended to make it clear in the manuscript.

There are many abbreviated proper nouns in the manuscript, such as NOVA, HEP, etc. The full name (abbreviation) should be used when it appears for the first time.

Figure 2, Figure 3a and Figure 3b are all very blurry and unclear. Please improve them.

Author Response

Dear Reviewer,

Thank you very much for taking the time to review this manuscript. Please find the detailed responses below and the corresponding revisions highlighted in the re-submitted files.

1. This review manuscript focuses on obesity and points out that in response to the growing problem, countries are introducing various dietary patterns to educate the public and promote healthy habits. Although in recent years, intuitive plate models have been used to replace the food pyramid. However, this manuscript only cites the United States' "My Plate", Harvard University's "Healthy Eating Plate" and Poland's "Healthy Eating Plate". Is there no relevant information for reference or comparison in other countries?

Thank you for your suggestions. In accordance with the reviewers' suggestions, the character of the article was changed to an original article. Two more dietary models were added to the comparative analysis: UK's The Eatwell Guide (UK, 2016) and Malaysian Healthy Plate (Malaysia, 2016). Those two were chosen because they are well described in the literature both in terms of structure and have scientific research and are published in English.

2. The United States launched "My Plate" and Harvard University's "Healthy Eating Plate" not only to improve the public's obesity problem, but also to consider other aspects of healthy eating, nutritional balance and disease prevention. It is recommended to make it clear in the manuscript.

Thank you for your suggestion. In our manuscript, the discussion section, expanded after reviews, describes the impact of using nutrition education tools not only on weight change, but also on changing eating habits and consequently improving lipid profiles, carbohydrate metabolism, reducing mortality. In the introduction, special attention was paid to the problem of obesity, as it is one of the biggest health problems in the world due to dietary errors, which could be reduced by proper nutrition education.

3. There are many abbreviated proper nouns in the manuscript, such as NOVA, HEP, etc. The full name (abbreviation) should be used when it appears for the first time.

Thank you for your remark. This has been corrected, only in the case of NOVA it remained unchanged, as it is not an acronym or abbreviation to be expanded, but a proper name.

4. Figure 2, Figure 3a and Figure 3b are all very blurry and unclear. Please improve them.

Thank you for your attention. The graphics have been enlarged and should now be clearer.

Best regards

Authors

Reviewer 4 Report

Comments and Suggestions for Authors

paper title - Comparative analysis of selected graphic models as a tool for nutrition education in the world with an assessment of their effectiveness in the context of the global obesity epidemic

here are some comments, suggestions and questions
in the abstract - should note what analyses are made,
only using literature review? - there should be some framework or rubic used in the comparison
or factors / dimensions used as criteria

introduction - 
Obesity is a chronic disease , which does not go away on its own and tends ... this should be cited
untreated obesity disease leads.....this should also be cited

should provide some more information or comparative perspective of obesity as compare to the study locale
sort of justify the significance of the study

lines 56 to 65 - no citations?
was not able to follow the succeeding paragraph  69 to 77
author/s should clarify more on this

should provide a method section - should describe how review were made, how many studies, as noted earlier there should be some sort of rubric or guidelines used as basis for comparison

Author Response

Dear Reviewer,

Thank you very much for taking the time to review this manuscript. Please find the detailed responses below and the corresponding revisions highlighted in the re-submitted files.

1. in the abstract - should note what analyses are made,

only using literature review? - there should be some framework or rubic used in the comparison

or factors / dimensions used as criteria

In accordance with the reviewers' suggestions, the character of the article was changed to an original article. The abstract has been revised to include the above suggestion.

introduction -

Obesity is a chronic disease , which does not go away on its own and tends ... this should be cited

untreated obesity disease leads.....this should also be cited

Thank you for your suggestions. Citations have been added.

2. should provide some more information or comparative perspective of obesity as compare to the study locale

sort of justify the significance of the study

Thank you for your remark. Obesity is only the background of our article, and in the introduction we have cited data from a report by the Supreme Audit Office on overweight and obesity in Poland.

Additionally, the significance was included in the purpose of the study and it was added to the introduction: “Therefore, the purpose of this study was to compare scientifically evaluated models of dietary recommendations from different countries with the Polish nutrition model in order to identify similarities and differences and potential areas for modification, and to propose directions for improvement based on research results.”

3. lines 56 to 65 - no citations?

Thank you for your attention. This was a writing error and has been corrected.

4. was not able to follow the succeeding paragraph  69 to 77

author/s should clarify more on this

Thank you for your comment. The text has been modified to make it clearer.

5. should provide a method section - should describe how review were made, how many studies, as noted earlier there should be some sort of rubric or guidelines used as basis for comparison

We have reclassified the manuscript as an original article and added a Materials and Methods section where this is described.

Best regards

Authors

Round 2

Reviewer 1 Report

Comments and Suggestions for Authors

Thank you to the authors for this revised version of the manuscript. This version shows substantial improvements. However, the methodology section still requires revision. First, it is unclear why a literature review is planned to select the figures, if the figures have already been chosen a priori based on their similarity to the image used in Poland. Please clarify the purpose of the literature review, especially considering that the figures were sourced from government agencies. If the literature review is still to be conducted, the authors should clearly explain its purpose and ensure it aligns with the study’s objectives.
Additionally, it would be important to clarify which limitations the authors are referring to—whether these relate to the study itself or to other aspects. If the limitations refer to the study itself, they should be included in the Discussion section rather than in the methodology.

Author Response

Dear Reviewer,

Thank you very much for taking the time to review this manuscript. Please find the detailed responses below and the corresponding revisions highlighted in the re-submitted files.

  1. Thank you to the authors for this revised version of the manuscript. This version shows substantial improvements. However, the methodology section still requires revision. First, it is unclear why a literature review is planned to select the figures, if the figures have already been chosen a priori based on their similarity to the image used in Poland. Please clarify the purpose of the literature review, especially considering that the figures were sourced from government agencies. If the literature review is still to be conducted, the authors should clearly explain its purpose and ensure it aligns with the study’s objectives.

Thank you for your kind assessment. The methodology section has been already revised  according to the other reviewers suggestions and uploaded. Your concerns are similar to those of other reviewers and are contained in the revisions.

  1. Additionally, it would be important to clarify which limitations the authors are referring to—whether these relate to the study itself or to other aspects. If the limitations refer to the study itself, they should be included in the Discussion section rather than in the methodology.

Thank you for your suggestion. The limitations of the study relate to the method used to conduct the study, including the availability of scientific literature on the subject, for which reason it is included in the Material and Methods section rather than the Discussion.

Best regards

Authors

Reviewer 2 Report

Comments and Suggestions for Authors

The manuscript was greatly improved following the first round of review, my only suggestion is to separate the section Materials and Methods in two parts: literature review for selection of graphic models for analysis; and qualitative analysis of graphic models.

Author Response

Dear Reviewer,

Thank you very much for taking the time to review this manuscript. Please find the detailed responses below and the corresponding revisions highlighted in the re-submitted files.

The manuscript was greatly improved following the first round of review, my only suggestion is to separate the section Materials and Methods in two parts: literature review for selection of graphic models for analysis; and qualitative analysis of graphic models.

Thank you for your kind assessment. The suggestion has been addressed in the Materials and Methods section.

Best regards

Authors

Reviewer 3 Report

Comments and Suggestions for Authors

The authors had provided additional information and responded in detail to my comments and suggestions.

Since most of the Graphic Models for Enhancing Nutrition Education in various countries are derived from the Food (Guide) Pyramid. However, the outdated daily dietary guidelines cannot accurately reflect the current dietary patterns of various countries and the need to improve obesity problems. Therefore, many countries continue to improve old dietary guidelines and develop dietary recommendations to improve current obesity and chronic disease problems.

It is recommended to appropriately supplement the above information in the introduction content.

Author Response

Dear Reviewer,

Thank you very much for taking the time to review this manuscript. Please find the detailed responses below and the corresponding revisions highlighted in the re-submitted files.

  1. The authors had provided additional information and responded in detail to my comments and suggestions.

Thank you for your kind assessment.

  1. Since most of the Graphic Models for Enhancing Nutrition Education in various countries are derived from the Food (Guide) Pyramid. However, the outdated daily dietary guidelines cannot accurately reflect the current dietary patterns of various countries and the need to improve obesity problems. Therefore, many countries continue to improve old dietary guidelines and develop dietary recommendations to improve current obesity and chronic disease problems. It is recommended to appropriately supplement the above information in the introduction content.

Thank you for your suggestion. This information has been added to the introduction section.

Best regards

Authors

Reviewer 4 Report

Comments and Suggestions for Authors

first would like to thanks the author for making the effort to enhance the paper  - however there are still some issues that needs to be revise

- lines 58 to 70 - only from 1 source?

- line 71.... transformation of ... this should also be cited

- shift not only enhance..... how did you tell - there should be citing literature

in the data analysis procedure - descriptive analysis seems to be more of a quantitative in nature - and this should be cited, 

would probably go with the initial notion of comparative analysis in line 85

however, the focus on structural differences and the amount of descriptive data used as a form of analysis should be elaborated - provide previous studies or some sort of model or framework to justify this mean of comparison

how many were initially selected .... how many excluded, since you have an inclusion and exclusion criteria

how many evaluated the data or did the comparative analysis - expert validity?

Author Response

Dear Reviewer,

Thank you very much for taking the time to review this manuscript. Please find the detailed responses below and the corresponding revisions highlighted in the re-submitted files.

  1. first would like to thanks the author for making the effort to enhance the paper  - however there are still some issues that needs to be revise

- lines 58 to 70 - only from 1 source?

Thank you for this question. Yes, the source of information in this paragraph is the summary of dietary recommendations in Europe prepared by the European Commission. The volume of the paragraph seems long relative to a single source, as it includes an enumeration of many European countries.

  1. - line 71.... transformation of ... this should also be cited

Thanks for the suggestion. Citations have been added.

  1. - shift not only enhance..... how did you tell - there should be citing literature

Thanks for the suggestion. Citation has been added.

  1. in the data analysis procedure - descriptive analysis seems to be more of a quantitative in nature - and this should be cited, 

At the suggestion of another reviewer, we modified the structure of the Materials and Methods section in that we removed the data analysis item.

  1. would probably go with the initial notion of comparative analysis in line 85,

however, the focus on structural differences and the amount of descriptive data used as a form of analysis should be elaborated - provide previous studies or some sort of model or framework to justify this mean of comparison

Thank you for your suggestion. A note has been added to the Materials and Methods section: “In light of the absence of prior studies comparing the models, a comprehensive evaluation of their graphical and descriptive components was conducted, with the aim of identifying similarities and differences and contextualizing these findings within the literature on recommendation effectiveness.”

  1. how many were initially selected .... how many excluded, since you have an inclusion and exclusion criteria

how many evaluated the data or did the comparative analysis - expert validity?

Thank you for your suggestion. A note has been added to the Materials and Methods section: “The literature review was conducted by two experts. Initially, 29 literature items found by keywords that matched the topic of the paper and complied with the inclusion criteria were included in the review. After careful analysis, 2 publications were rejected due to inconsistency with the others.”

Best regards

Authors

Round 3

Reviewer 4 Report

Comments and Suggestions for Authors

after going over the revisions made by the authors, i think the paper is now appropriate for acceptance